# Novel Nile Blue Analogue Stains Yeast Vacuolar Membrane, Endoplasmic Reticulum, and Lipid Droplets, Inducing Cell Death through Vacuole Membrane Permeabilization

**DOI:** 10.3390/jof7110971

**Published:** 2021-11-16

**Authors:** João Carlos Canossa Ferreira, Carla Lopes, Ana Preto, Maria Sameiro Torres Gonçalves, Maria João Sousa

**Affiliations:** 1Centre of Molecular and Environmental Biology (CBMA), Department of Biology, Campus of Gualtar, University of Minho, 4710-057 Braga, Portugal; carla.lopes89@hotmail.com (C.L.); apreto@bio.uminho.pt (A.P.); 2Campus of Gualtar, IBS-Institute of Science and Innovation for Bio-Sustainability, University of Minho, 4710-057 Braga, Portugal; 3Centre of Chemistry, Department of Chemistry, Campus of Gualtar, University of Minho, 4710-057 Braga, Portugal; msameiro@quimica.uminho.pt

**Keywords:** benzo[*a*]phenoxazine derivative, Nile Blue analogue, cell death mechanism, vacuole/lysosome membrane permeabilization, yeast as a eukaryotic cell model

## Abstract

Phenoxazine derivatives such as Nile Blue analogues are assumed to be increasingly relevant in cell biology due to their fluorescence staining capabilities and antifungal and anticancer activities. However, the mechanisms underlying their effects remain poorly elucidated. Using *S. cerevisiae* as a eukaryotic model, we found that BaP1, a novel 5- and 9-*N*-substituted benzo[*a*]phenoxazine synthesized in our laboratory, when used in low concentrations, accumulates and stains the vacuolar membrane and the endoplasmic reticulum. In contrast, at higher concentrations, BaP1 stains lipid droplets and induces a regulated cell death process mediated by vacuolar membrane permeabilization. BaP1 also induced mitochondrial fragmentation and depolarization but did not lead to ROS accumulation, changes in intracellular Ca^2+^, or loss of plasma membrane integrity. Additionally, our results show that the cell death process is dependent on the vacuolar protease Pep4p and that the vacuole permeabilization results in its translocation from the vacuole to the cytosol. In addition, although nucleic acids are commonly described as targets of benzo[*a*]phenoxazines, we did not find any alterations at the DNA level. Our observations highlight BaP1 as a promising molecule for pharmacological application, using vacuole membrane permeabilization as a targeted approach.

## 1. Introduction

Phenoxazine derivatives are long wavelength fluorescent dyes of great importance as probes since they absorb and emit fluorescence in the 600–900 nm region of the spectrum where there is minimum interference with the biological molecules’ auto-fluorescence [1,2,3,4,5,6]. They have been used in the covalent labeling of amino acids and proteins and in the non-covalent labeling of nucleic acids in various contexts such as blotting experiments, gel electrophoresis, and living cell assays [7,8,9,10,11].

In addition to their use as fluorophores, oxazine heterocycles such as phenoxazine and benzo[*a*]phenoxazine derivatives have also assumed significance in life sciences due to their antiproliferative proprieties, which has prompted their study in pharmacological applications such as antimicrobial [12,13,14] and antitumor agents [15,16]. These compounds may exert their cytotoxicity through the intercalation of their planar structure between the DNA base pairs, forming stable complexes. In addition, they can suffer a metabolic conversion into free radical intermediates leading to oxidative stress [17]. Some phenoxazine derivatives displayed apoptotic activity in different neoplastic cell lines, both in a caspase-dependent and independent manner [18,19,20,21,22], while others induced autophagic cell death in apoptosis-resistant cells [23]. This class of compounds has been shown to accumulate in cancer cells to a higher extent than in normal human cells, resulting in increased toxicity toward cancer cells [24,25]. Despite their interest, information regarding their use and mechanisms of action is very limited and further research is needed to explore their potential. In recent years, our group has synthesized several benzo[*a*]phenoxazine compounds with significant antiproliferative activity against *Saccharomyces cerevisiae*. Among these, the Nile Blue analogue *N*-(5-((4-ethoxy-4-oxobutyl)amino)-10-methyl-9*H*-benzo[*a*]phenoxazine-9-lidene) BaP1 (Figure 1) was particularly active [13,26,27].

In the present work, we show that in addition to its antiproliferative activity, BaP1 induces cell death in yeast cells and we were able to unveil the mechanism behind its cell death-inducing activity. Using *S. cerevisiae* as a model organism, we assessed BaP1 intracellular localization and cell death markers upon incubation with low doses and with a cytotoxic concentration of BaP1, respectively.

## 2. Materials and Methods

### 2.1. Yeast Strains

The following *Saccharomyces cerevisiae* strains were used in this study: BY4741 (*MATa, his3*Δ*1 leu2* Δ*0 met15* Δ*0 ura3* Δ*0*), (EUROSCARF, Frankfurt, Germany), and the respective knockouts Δ*pep4*, Δ*nuc1,* Δ*aif1 and* Δ*cpr3*; BY4741 SEC66-GFP (derived from the *S. cerevisiae* BY4741), as well as BY4741 *Δpep4* constructions (BY4741 Δ*pep4* pESC(Ø), BY 4741 Δ*pep4* pESC-Pep4p(FL) and BY 4741 Δ*pep4* pESC-DPM-Pep4p); BY4742 (*MATa his3* Δ*1 leu2* Δ*0 lys2* Δ*0 ura3* Δ*0*) (EUROSCARF, Frankfurt, Germany), and the respective knockouts Δ*fld1,* Δ*tgl3* Δ*tgl44,* Δ*are1* Δ*are2,* Δ*dga1* Δiro1; W303-1A (*MATa*, *ura3-52, trp1* Δ*2, leu2-3, 112 his3-11, ade2-1*), (EUROSCARF, Frankfurt, Germany), and the respective mutants *rho*^0^, Δ*aac 1/2/3,* Δ*atg1*, Δ*yca*, as well as the constructions W303-1 pDF01-VBA1-YEGFP, W303-1A pYX-mt-GFP, W303-1A p416 ADH-pep4-EGFP, W303-1A GFP-ATG8, W303-1A Δ*atg5* GFP-ATG8 and W303-1A Δ*atg32* GFP-ATG8; US50-18 C (*ura3, his1, pdr1-3*) and the knockout mutant AD1-7 (*ura3, his1, pdr1-3,* Δ*yor1,* Δ*snq2,* Δ*pdr5,* Δ*pdr10,* Δ*pdr11,* Δ*ycf1,* Δ*pdr3*).

### 2.2. Growth

Yeast cells were grown on standard YEPD (1% yeast extract, 2% peptone, 2% glucose). The strains transformed with plasmids were selected and grown in synthetic complete medium (SC: 2% glucose, 0.5% ammonium sulfate, 0.7% yeast nitrogen base w/o amino acids, 0.2% dropout mix, 0.01% histidine, uracil, and tryptophan, 0.02% leucine). In the case of BY 4741 Δ*pep4* (pESC(Ø), pESC-Pep4p(FL) and pESC-DPM-Pep4p) and W303-1A pYX-mt-GFP strains, to induce protein expression, the same medium with galactose 2%, instead of glucose, was used. All the strains were grown at 30 °C with agitation at 200 rpm, until they reached the early exponential phase (OD_640 nm_ ≈ 0.5). For solid media, 2% agar was added.

### 2.3. Survival Assays

When yeast cultures (cultivated as described above) reached the early exponential phase (OD_640 nm_ ≈ 0.5), cells were collected, washed, and resuspended in filtered YEPD or YEPG (in the case of strains with Gal promotor). The treatment with the compound was performed by adding BaP1 to the resuspended cells at a final concentration of 300 μM. An 80.0 mM stock solution of BaP1 in dimethyl sulfoxide (DMSO) was prepared. The purity of the synthesized BaP1 compound was determined by ^1^H RMN analysis and was higher than 95% [26,27]. Since BaP1 was dissolved on DMSO, the same volume of DMSO (≈0.35%) was added to another tube that served as the negative control. The cell suspensions were incubated at 30 °C with agitation at 200 rpm for 120 min, and 50 μL samples were collected every 30 min. The 0 min sample was collected before adding the compound or DMSO (negative control). Samples were serially diluted to 10^−4^ in sterile deionized water. Aliquots of the 10^−4^ dilution were placed on YEPD plates for assessing colony forming units (CFUs). The plates were incubated during 2 days at 30 °C and CFUs were counted. For protein synthesis inhibition assays, cycloheximide (CHX) (Sigma-Aldrich) was dissolved in ethanol and added to yeast cell cultures grown to (OD_640 nm_ ≈ 0.5) at a final concentration of 100 μg/mL. The cultures were incubated for 30 min before BaP1 treatment.

### 2.4. Cell Death Markers

Plasma membrane integrity was accessed by incubating the cells for 5 min with 2 μg/mL of propidium iodide (PI) (Molecular Probes, Eugene, OR, USA) at room temperature followed by the flow cytometry measurements of PI-stained cells. Boiled cells were used as positive control.

For the assessment of mitochondrial potential, DiOC_6_(3) (Molecular Probes, Eugene, OR, USA) was used. Cells were collected, washed in PBS, and resuspended in DiOC buffer (0.1 mM MgCl_2_, 10 mM MES (2-(*N*-morpholino)ethanesulfonic acid) and 2% (*w/v*) glucose, pH 6.0). Then, they were incubated with DiOC_6_(3) at a final concentration of 0.24 μM for 30 min at 30 °C in the dark. The samples were analyzed by flow cytometry. As a positive control, cells were treated with 150 mM acetic acid.

Intracellular reactive oxygen species were detected by 2′,7′-dichlorodihydrofluorescein diacetate (H_2_DCFDA). For the detection of ROS, a double staining protocol with H_2_DCFDA and PI was used. Cells were collected and then resuspended in culture medium containing 40 μg/mL H_2_DCFDA at 30 °C for 45 min, in the dark and then 2 μg/mL of PI was added after 30 min. Cells were analyzed by flow cytometry.

For intracellular calcium measurements, cells previously washed with PBS were stained with 5 μM FLuo4-AM (Molecular Probes, Eugene, OR, USA) for 90 min at 30 °C in the dark, subsequently washed and resuspended in PBS and assessed by flow cytometry. As the positive control, cells were treated with 3 mM H_2_O_2_ at pH 3.0 [28].

DNA strand breaks were assessed by TUNEL with the “In Situ Cell Death Detection Kit, Fluorescein” (Roche Applied Science) as previously described in [29]. To measure DNA content, cells were stained with SYBR Green I as described in [30] and staining was assessed by flow cytometry. The quantification of the different cell cycle phases was performed by the evaluation of the DNA content through the quantification of the Fl-1 fluorescent intensity.

Flow cytometry was performed in an Epics^®^ XL™ (Beckman Coulter, Pasadena, CA, USA) flow cytometer, equipped with an argon-ion laser emitting a 488 nm beam at 15 mW. The monoparametric detection of PI fluorescence was performed using FL-3 (488/620 nm) and the detection of DiOC_6_(3), H_2_DCFDA, FLuo4-AM and SYTOX Green fluorescence was performed using Fl-1 (488/525 nm). Thirty thousand cells were analyzed per sample at a low flow rate. Flow cytometry analyses were performed with FlowJo^®^ 7.6 Software, Ashland, OR, USA.

### 2.5. Detection of Autophagy Induction in Yeast

Autophagy induction was accessed through the intracellular localization of the GFP-ATG8 protein by fluorescence microscopy and by Western blot evaluation of GFP-ATG8 processing. As the positive control, cells were treated with the autophagy inducer rapamycin at a final concentration of 0.2 μg/mL for 60 min. Western blot analysis was performed according to [31].

For GFP and Pgk1p detection, a mouse anti-GFP monoclonal primary antibody (1:2000; Roche, Mannheim, Germany) and mouse monoclonal anti-yeast phosphoglycerate kinase (PGK1) antibody (1:5000; Invitrogen, Waltham, MA, USA) were used. As the secondary antibody, a mouse IgG coupled to horseradish peroxidase (HRP) (1:5000; Jackson ImmunoResearch, West Grove, PA, USA) was used. The signals were revealed by chemiluminescence detection using the Immobilon Western HRP Substrate (Millipore-Merck, Darmstadt, Germany) and a ChemiDoc XRS image system (BioRad, Hercules, CA, USA).

### 2.6. Evaluation of BaP1 Intracellular Distribution and Vacuole Permeabilization by Fluorescence Microscopy

For fluorescence microscopy observations, the cells were collected at different time points, washed, and mounted on a slide.

BY4741, SEC66-GFP, W303-1 pDF01-VBA1-YEGFP, and BY4742, and the respective knockouts Δ*fld1,* Δ*tgl3* Δ*tgl44,* Δ*are1* Δ*are2,* and Δ*dga1* Δ*iro1* were used to determine the intracellular distribution of BaP1. For the mitochondria morphology observations, the W303-1A pYX-mt-GFP strain was used. Vacuolar membrane permeabilization was analyzed by CMAC staining, BY 4741 cells were stained with Celltracker™ Blue CMAC (Molecular Probes, Eugene, OR, USA) at a final concentration of 2 μM and incubated for 15–30 min at room temperature. The release of Pep4p to the cytosol was analyzed using W303-1A p416 ADH-pep4-EGFP.

The samples were observed on a Leica MicrosystemsDM-5000B microscope with appropriate filter settings (red, green, blue, and DIC (Differential Interference Contrast)) and a 100× oil immersion objective. Images were obtained with a Leica DFC350 FX Digital Camera and processed with LAS AF Microsystems software. For phenotype quantification, at least 300 cells of at least two independent experiments were evaluated.

## 3. Results

### 3.1. BaP1 Causes Viability Loss in Yeast Cells Partially Dependent on Protein Synthesis

In a previous report [13], we showed that BaP1 inhibited yeast proliferation. To assess whether BaP1 could induce cell death, *S. cerevisiae* cells were incubated with this compound and its effects on cell viability were determined by CFU counting over time. We observed that BaP1 induced a fast decrease in cell viability in *S. cerevisiae,* the effect being partially reverted by the addition of the protein synthesis inhibitor cycloheximide [29] (Figure 2A).

### 3.2. Yeast PDR Transporters Do Not Confer Resistance to BaP1

Drug export is a general mechanism of multidrug resistance. In *S. cerevisiae*, a complex network of genes, the pleiotropic drug resistance (PDR) network confers resistance to a variety of small cytotoxic molecules by activating their cellular efflux through ABC efflux pumps [32,33]. To understand whether the toxic effect of BaP1 could be reduced by its efflux out of the cell, we evaluated the effect of the compound on cell viability of a PDR-efflux deleted mutant, AD1-7 (deleted in six genes encoding the transporters Yor1p, Snq2p, Pdr5p, Pdr10p, Pdr11p, Ycf1p, and the transcription factor Pdr3p), in contrast to US50-18C, a strain with the activating mutation *pdr1–3* in the gene encoding the transcription factor Pdr1p, hyper-resistant to drugs which are PDR substrates [34]. The viability assay showed that US50-18C was more sensitive to the action of BaP1 while AD1–7 was more resistant to its effect (Figure 2B). This observation shows that BaP1 is not a substrate of any of the tested drug–efflux pumps belonging to the ABC family of transporters and suggests that changes in membrane lipid composition may influence BaP1 activity, as these ABC family proteins have been implicated in the modulation of the lipid composition of *S. cerevisiae* membranes through the activation of sphingolipid biosynthesis along with other target genes [35].

### 3.3. BaP1-Induced Cell Death Is Dependent on Vacuolar Protease Pep4p; However, Not on Commonly Described Mitochondrial Apoptotic Regulators

The observation that BaP1 was able to induce cell death, partially reduced by the inhibition of the protein synthesis, suggested the regulated nature of the process, and led us to assess the potential role of proteins reported to be involved in apoptotic or necrotic cell death. As such, we tested yeast strains lacking: metacaspase (Δ*yca1*), ADP/ATP carrier of the mitochondrial inner membrane (Δ*aac1/2/3*), vacuolar protease Pep4p (Δ*pep4*), nuclease Nuc1p, described to translocate from mitochondria to the nucleus upon an apoptotic stimulus (Δ*nuc1*) and apoptosis-inducing factor Aif1p (Δ*aif1*) which translocates to the nucleus in response to apoptotic stimuli. We also tested the Δ*cpr3* strain, deficient in the gene coding for mitochondrial cyclophilin, the yeast orthologue of mammalian cyclophilin D, described as a mediator of necrosis, and a strain without mitochondrial DNA, *rho*0 [36,37,38,39]. Viability assays showed that Δ*aac1/2/3* and *rho*0 were slightly more sensitive and Δ*yca1* was more resistant to the action of BaP1. However, despite being consistent, these differences were not statically significant (Figure 2C). The mutant strains Δ*nuc1*, Δ*aif1,* and Δ*cpr3* showed no differences when compared with the wild-type strain (Figure 2C,D).

On the other hand, the strain lacking the vacuolar protease Pep4p was considerably more resistant to the effect of BaP1 (Figure 2D). Pep4p is a pepsin-like aspartic protease orthologue of human cathepsin D that was shown to translocate from the yeast vacuole to the cytosol during hydrogen peroxide [40] and acetic acid-induced regulated cell death, also being involved in mitochondrial degradation and cell death protection during the latter process [41]. The fact that the Δ*pep4* strain is more resistant to the BaP1 effect suggests that this protein plays a role in the mediation of the cell death process. To confirm the effect of Pep4p in the cell death process and to investigate whether this resistance relies on the proteolytic activity of the protein, *S. cerevisiae* BY4741 Δ*pep4* strains transformed either with pESC-Pep4p(FL) containing a functional Pep4p, the double point mutation, pESC-DPM-Pep4p proteolytic inactive, or the empty plasmid pESC (Ø), were used in viability assays. As expected, BY4741 Δ*pep4* pESC(Ø) showed a reduced loss of viability, similarly to the BY4741 Δ*pep4* strain (Figure 2E). The resistance phenotype was reverted by the expression of the functional Pep4p, Pep4p(FL) strain displaying a significant difference from the strain containing the empty plasmid in the last time points of the assay (Figure 2E). Furthermore, the proteolytic activity of Pep4p appears to be relevant for cell death mediation, since DPM-Pep4p presents a lower viability loss than Pep4p(FL) (Figure 2E). These data confirm that Pep4p plays an important role in the cell death process.

### 3.4. BaP1 Leads to Mitochondrial Depolarization and Fragmentation with Preserved Plasma Membrane Integrity

To characterize the cell death process induced by BaP1, apoptotic or necrotic markers/parameters were analyzed along the treatment time, namely: plasma membrane integrity, mitochondrial fragmentation, mitochondrial membrane potential alterations, reactive oxygen species (ROS) accumulation, intracellular calcium fluctuations, and DNA alterations.

Mitochondria are dynamic organelles continuously dividing and fusing to form an active interconnecting network [42]. However, during apoptosis, this network disintegrates at the time of cytochrome *c* release and prior to caspase activation, yielding more numerous and smaller mitochondria [43]. Therefore, we assessed whether BaP1 led to mitochondria network fragmentation using a *S. cerevisiae* strain expressing a mitochondrial-targeted GFP protein. When treated with BaP1, yeast cells revealed the appearance of punctuated structures instead of a defined interconnected network (Figure 3A). After 30 min of exposure, more than 80% of cells displayed mitochondrial network fragmentation and the value increased up to almost 92% after 120 min of treatment (Figure 3B).

Several studies have shown that alterations in the mitochondrial membrane potential are implicated in the occurrence of regulated cell death. Thus, we evaluated whether BaP1 induces changes in the mitochondrial membrane potential. As such, we used the carbocyanine dye DiOC_6_(3) which accumulates in mitochondria depending on its membrane potential. As a positive control, cells were treated with acetic acid, as previous work showed a transient hyperpolarization followed by the depolarization of mitochondrial membrane during acetic acid regulated cell death [41]. In agreement with what was described for the acetic acid-treated cells, we observed a shift of the DiOC_6_(3) fluorescence peak to the left, showing mitochondrial depolarization, and a small peak in the right side of the histogram, which corresponds to cells with mitochondrial hyperpolarization (Appendix A), In the BaP1-treated cells, we could not observe any shift in the DiOC_6_(3) main peak. However, we observed the appearance of an additional peak in the left of the histogram (Appendix A), which shows the existence of a sub-population presenting mitochondrial depolarization. We could observe that, after 30 min of treatment, approximately 18% of the total cells presented mitochondrial depolarization; however, this value did not increase over 23%, even in the last time points (Figure 3C).

Considering the fact that mitochondria can represent a source of ROS production within cells [44,45] and that phenoxazine derivatives may lead to oxidative stress [17], we assessed whether BaP1 induced ROS accumulation using a double staining protocol with propidium iodide (PI) and H_2_DCFDA. However, we found that 300 μM of BaP1 did not lead to an increased accumulation of ROS (Appendix A).

Intracellular calcium perturbations have been reported as a cause of cytotoxicity and apoptotic or necrotic cell death [46]. However, using Fluo-4, a probe to detect intracellular Ca^2+^, no alterations in calcium levels were observed (Appendix A).

In apoptotic cell death, cells are characterized by alterations at the DNA level, such as fragmentation, resulting in the appearance of DNA strand breaks and diminished DNA content, characterized by the appearance of a sub-G0/G1 peak in a cell cycle analysis histogram [47,48]. We assessed chromatin fragmentation by TUNEL assay and cell DNA content by flow cytometry using SYTOX Green staining. Results showed that less than 1% of treated cells presented DNA fragmentation, even at the last time points (Appendix A). Furthermore, the SYTOX Green staining showed no significant change in the number of cells in the sub G0/G1 phase (Appendix A).

The assessment of the plasma membrane integrity of BaP1 treated cells by flow cytometry using PI staining showed that the percentage of PI-positive cells did not increase significantly, not exceeding 1% (Appendix A), confirming that this is not a necrotic process. 

Taken together, these observations indicate that BaP1 induces cell death with the preservation of plasma membrane integrity, mitochondrial fragmentation, and depolarization, but lacks some other common apoptotic markers.

### 3.5. Autophagy Is Not Involved in the BaP1 Cell Death

Since autophagic cell death induction by benzo[*a*]phenoxazine derivatives was recently reported [23], we questioned the potential involvement of the autophagy machinery in the cell death induced by BaP1. To address this question, we tested the dependence of the cell death process on Atg1p, a serine-threonine kinase that is involved in the formation of autophagosome membranes which is essential for autophagy induction [49]. No differences between the wild-type and Δ*atg1* strains (Figure 4A) were observed in cell viability assays, indicating that Atg1p, and thus autophagic induction, does not have a role in BaP1-induced cell death.

Another assay that is commonly used to detect autophagy occurrence is the processing of a GFP-ATG8 fusion protein [50]. During the course of autophagy, Atg8p is incorporated into the autophagosome and carried to the vacuolar lumen where it is degraded by the vacuolar hydrolases [51,52]; the hydrolysis of GFP-ATG8 results in free GFP that can be detected as an autophagy indicator. Considering this approach, we used W303-1A GFP-ATG8 cells, which were treated with 300 μM BaP1 or with 0.2 μg/mL rapamycin, an autophagy inducer used as the positive control. In contrast to rapamycin treatment, with BaP1 treatment, no particular fluorescence in the vacuolar lumen was detected (Figure 4B). Moreover, we could not detect any increase in the cleaved GFP-ATG8 band by Western blot analysis, so the occurrence of autophagy seems very unlikely (Figure 4C,D). As expected, in an autophagy-deficient strain, Δ*atg5,* green fluorescence remained dispersed in the cytosol for both treatments, and no cleaved band was observed, confirming that the rapamycin phenotype observed was dependent on autophagy (Figure 4A–D).

### 3.6. BaP1 Accumulates at the Vacuolar Membrane, Endoplasmic Reticulum, and Lipid Droplets

In order to analyze BaP1 fluorescence staining and to obtain further insights into its possible intracellular targets, we used fluorescence microscopy to assess BaP1 intracellular distribution. As such, we co-incubated *S. cerevisiae* cells with BaP1 and organelle-specific fluorescent probes, or used cells expressing GFP-fusion proteins targeting specific organelles such as mitochondria, vacuole, endoplasmic reticulum, endosomes, Golgi complex, and nuclei. Since BaP1 is highly fluorescent and to allow a better definition, in the staining experiments, we decreased the concentration of the compound to 2.5 µM. Using a strain expressing VBA1-YEGFP fusion protein, which allows the localization of the vacuolar membrane, we observed that the compound’s Far-Red fluorescence co-localized with the fusion protein. (Figure 5A). In addition, using a strain expressing a GFP fusion of the endoplasmic reticulum protein Sec66, we could observe that BaP1 also localizes at the perinuclear endoplasmic reticulum (ER) since the colocalization of the Far-Red and green fluorescence was observed (Figure 5B). We could not observe any significant co-staining with mitochondria, endosomes, Golgi complex, or nuclei (not shown).

Interestingly, at 300 μM, the concentration of the compound used to induce a loss of cell viability in *S. cerevisiae*, BaP1 loses its Far-Red vacuolar and ER staining specificity, presenting a more diffuse staining pattern. However, we observed the appearance of punctuated structures in the cytoplasm emitting green and blue fluorescence (Figure 5C). Since the benzophenoxazine family compounds, Nile Blue, and Nile Red, have affinity for lipophilic molecules [53] and particularly Nile Red has been used as a probe for lipid droplets (LDs) detection, we analyzed whether the green and blue fluorescence came from BaP1 LD staining. As such, we used *S. cerevisiae* mutants with different lipid droplet morphology and abundance and were able to confirm our hypothesis (Appendix A).

BaP1 accumulation in LDs does not seem to influence cell death induced by the compound, as cell viability assays performed with the BY4742 *∆dga1∆iro1* strain, with much lower lipid droplet content, did not reveal differences to the wild type (Figure 2D).

### 3.7. BaP1 Leads to Vacuolar Membrane Permeabilization and Cytosolic Release of Pep4p

The release of vacuolar and/or lysosomal contents to the cytosol has been observed to play a role in several cell death processes. Furthermore, it is known that the magnitude of the vacuolar content release determines the cell death type, a complete disruption of vacuole usually being associated with uncontrolled cell death by necrosis, whereas partial and selective permeabilization is associated with a controlled cell death process by apoptosis [54,55].

Considering the fact that BaP1 accumulates at the vacuolar membrane, we evaluated whether BaP1 could be disturbing the integrity of the vacuolar membrane during the cell death induction process. For this we used the fluorescent dye CMAC, which accumulates in the vacuolar lumen if the vacuolar membrane is intact. We could observe that, at 30 min of treatment with a cell death-inducing dose, the blue fluorescence present in the vacuolar lumen before BaP1 treatment (control) was already dispersed through the cell in more than 60% of the cells (Figure 6A,C) and that this value increased up to 90% after 120 min of treatment. These observations indicated the occurrence of vacuolar membrane permeabilization. Considering the Pep4p role in the cell death process, we assessed the occurrence of Pep4p translocation to the cytosol. Using a *S. cerevisiae* strain expressing a Pep4p-EGFP, we were able to observe green labeling throughout the cell, indicating the translocation of the protein to the cytosol already at 30 min of treatment (Figure 6B).

## 4. Discussion and Conclusions

In previous work in our laboratory, we found that several newly synthetized fluorescent compounds derived from benzo[*a*]phenoxazine displayed antiproliferative activity against *S. cerevisiae,* used as a model eukaryotic organism, and that minor changes in substituents of the benzo[*a*]phenoxazine nucleus drastically influenced their activity, with MIC values ranging from 3.75 μM to over 400 μM, as well as their intracellular staining patterns [7,13,14,26,27,56]. Here, we found that one of these compounds, BaP1, induced a regulated cell death process and uncovered its mode of action to unveil its potential for pharmacological applications.

Drug resistance is one of the first barriers found in the pharmacological application of many compounds. At this level, the most important resistance mechanism, ubiquitous from bacteria to humans, is the overexpression of membrane-associated transporters that extrude drugs out of the cell [57]. In *S. cerevisiae,* the active efflux of drugs is mediated by ABC efflux pumps that are regulated by a complex network of genes belonging to the pleiotropic drug resistance (PDR) network [32,33]. In this case, we showed that BaP1 cell death induction is not affected by the deletion or overexpression of PDR transporters, suggesting that BaP1 is not a substrate of these pumps being a positive indication for its application. Regarding the cell death process, we observed that BaP1 induced mitochondrial fragmentation and depolarization, with no loss of plasma membrane integrity. We further observed that BaP1 leads to vacuolar membrane permeabilization (VMP). The consequences of VMP are well documented in the literature in different scenarios. These organelles possess a large amount of proteases that, when in the cytosol, may be responsible for the activation of apoptotic effectors such as caspases and mitochondria that signal-regulated cell death [58,59,60,61]. When VMP is extensive, it can lead to widespread hydrolysis in the cytoplasm, causing necrosis [54]. Our results do not corroborate a mechanism of this type, as we clearly show that BaP1 does not induce a necrotic process. In fact, we observed that BaP1-induced VMP is accompanied by the release of Pep4p and that Pep4p-deficient cells are more resistant to BaP1-induced cell death, the results being in accordance with a pro-death role of Pep4p. An effector role of Pep4p has also been evidenced during H_2_O_2_-induced apoptosis, where Pep4p translocates from the vacuole to the cytosol and degrades the nucleoporins [40]. On the other hand, it has been shown that acetic acid can induce VMP with the release of Pep4p in yeast, which seems to have an anti-apoptotic effect, protecting cells from acetic acid-induced regulated cell death [41].

Our results also showed that the autophagic machinery was not involved in the cell death process. This result is in accordance with previous reports showing that VMP blocks autophagy and leads to autophagosome accumulation [59]. Our observations are also in agreement with those described by Suzuki et al. who showed that two phenoxazine derivatives, VM7 and VM8, inhibited autophagosome formation and induced cell death on tumor cells and normal human cells [20].

To the best of our knowledge, this is the first report in the literature regarding the involvement of VMP and Pep4p in a cell death process induced by a benzo[*a*]phenoxazine derivative. 

The assessment of BaP1 cellular distribution in low doses revealed that the compound stains the ER and the vacuolar membrane. In this context, BaP1 arises as a suitable molecule to be used as a fluorescent probe for ER and vacuole co-localization experiments. This is reinforced by BaP1’s great photophysical characteristics, such as its emission in the Far-Red region, where background interference caused by the biological material is minimal, as well as presenting high photostability, molar absorption, and modest stoke shift [26,27]. In contrast, at higher doses, capable of inducing loss of cell viability, BaP1 changes its staining profile as it loses its Far-Red vacuolar and ER staining specificity, probably due to the depolarization of the membranes [62], presenting a more diffuse staining pattern, accompanied by green and blue lipid droplets’ fluorescence stain. However, our results show that there is not a direct correlation between BaP1 cell death induction and its accumulation on LDs, as a yeast strain with reduced accumulation of LDs showed no differences in viability when compared with the wild-type strain. This is in accordance with other observations where compounds of this family were reported to stain these lipidic structures [63,64,65].

Overall, our data show the huge versatility of this compound. Its fluorescence staining pattern together with its photophysical properties highlight BaP1 as an endorsed fluorescent probe for biological applications.

On the other hand, BaP1 shows immense potential to be explored not only as an antifungal but also in other pharmacological applications. Phenoxazine compounds have grown as molecules of great interest in anticancer therapy [18,19,20,21,22,23,66] as well as antibacterial [67,68] and antiviral agents [69]. In this perspective, BaP1’s vacuolar membrane-permeabilizing capacity could emerge as a promising therapeutical approach.

## Figures and Tables

**Figure 1 jof-07-00971-f001:**
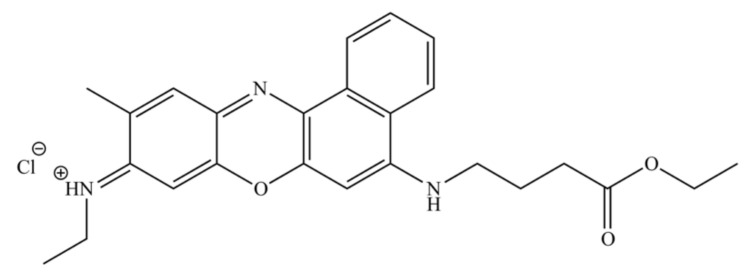
Chemical structure of *N*-(5-((4-ethoxy-4-oxobutyl)amino)-10-methyl-9*H*-benzo[*a*]phenoxazin-9-ylidene)ethanaminium chloride (BaP1).

**Figure 2 jof-07-00971-f002:**
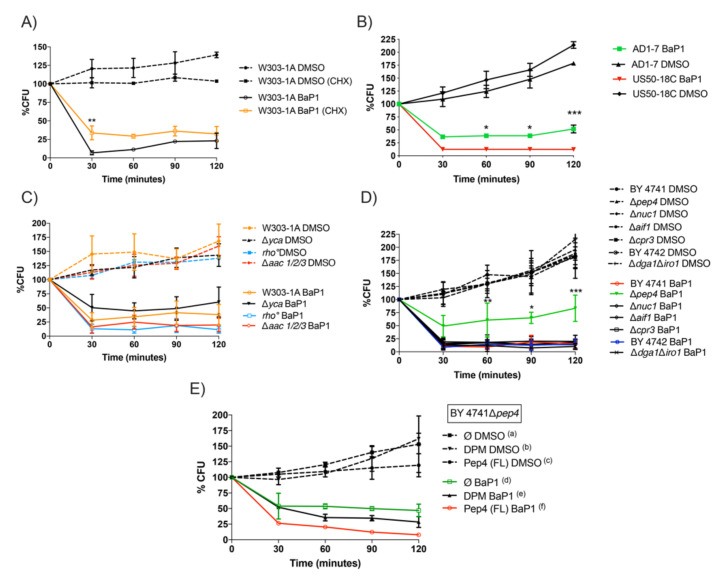
Effect of BaP1 on yeast cellular viability. (**A**) Effect of cycloheximide (CHX) (100 μg/mL) on viability of *S. cerevisiae* W303-1A exponential cells, exposed to BaP1 (300 μM) and DMSO (0.35%) (negative control). Cycloheximide was added 30 min before the beginning of BaP1 treatment. (**B**) Effect of BaP1 (300 μM) and DMSO (0.35%) on cell viability of *S. cerevisiae* AD1-7 and US50-18C strains. (**C**) Effect of BaP1 (300 μM) and DMSO (0.35%) on cell viability of *S. cerevisiae* W303-1A, Δ*yca1,* Δ*aac 1/2/3, rho*0. (**D**) Effect of BaP1 (300 μM) and DMSO (0.35%) on cell viability of *S. cerevisiae* BY4741, BY4742, Δ*pep4,* Δ*nuc1,* Δ*aif1,* Δ*cpr3, and* Δ*dga1∆iro1* strains. (**E**) Effect of BaP1 (300 μM) and DMSO (0.35%) on cell viability of *S. cerevisiae* BY 4741 Δ*pep4* pESC(Ø), BY 4741 Δ*pep4* pESC-DPM-Pep4p and BY 4741 Δ*pep4* pESC-Pep4p(FL) strains. 30 min: (d vs. e) ns, (d vs. f) **, (e vs. f) **; 60 min: (d vs. e) *, (d vs. f) ***, (e vs. f) ns; 90 min: (d vs. e) ns, (d vs. f) ***, (e vs. f) *; 120 min: (d vs. e) *, (d vs. f) ***, (e vs. f) *. Cell viability was assessed by CFU counting at the different time points (0, 30, 60, 90, 120 min). The time 0 corresponds to 100% is the CFU counting before BaP1 and DMSO treatment. Values are means with SD (*n* ≥ 2). Statistical analysis was performed by two-way ANOVA. * *p* < 0.05, ** *p* < 0.01, *** *p* < 0.001.

**Figure 3 jof-07-00971-f003:**
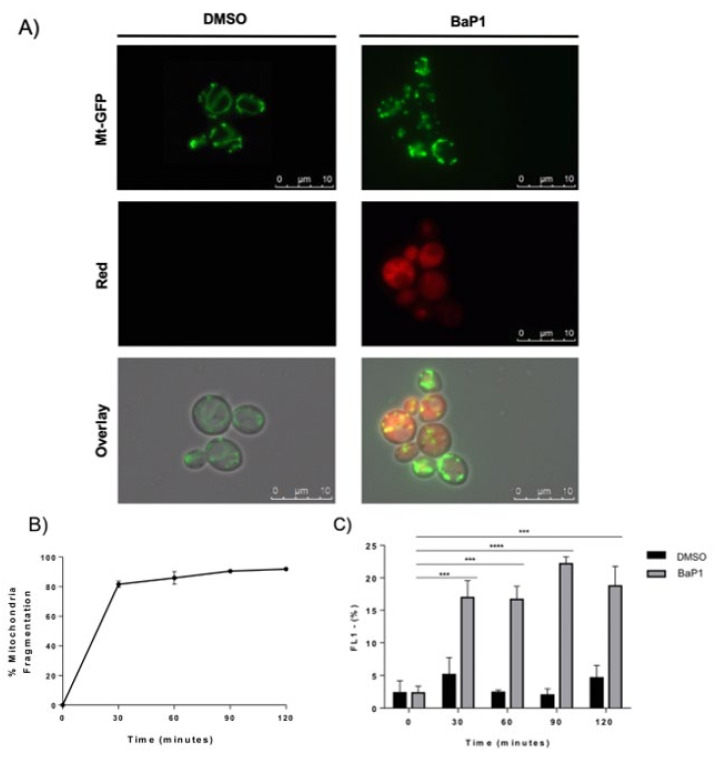
Evaluation of mitochondrial morphology and mitochondrial membrane potential. (**A**) Fluorescence microscopy images of W303-1A pYX-mt-GFP cells after 30 min of treatment with BaP1 (300 μM) or DMSO (negative control). Samples were collected at different time points, before (time 0) and after 30, 60, 90, and 120 min of treatment, and then visualized by epifluorescence microscopy with a 100× oil immersion objective; (**B**) percentage of BaP1-treated cells displaying mitochondrial network fragmentation. At least 300 cells were counted for each condition. (**C**) Effect of BaP1 on mitochondrial membrane potential. Quantification of FL1 negative population (FL1 LOG) corresponding to the percentage of cells with mitochondrial depolarization. Values are the means with SD (*n* ≥ 2). Statistical analysis was performed by two-way ANOVA. ns: non-significant, *** *p* < 0.001, **** *p* < 0.0001.

**Figure 4 jof-07-00971-f004:**
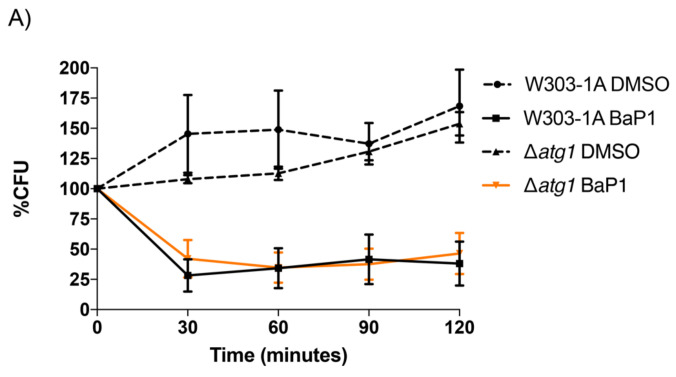
Evaluation of autophagy induction. (**A**) Effect of BaP1 (300 μM) and DMSO (0.35%) on cell viability of *S. cerevisiae* W303-1A and W303-1A ∆*atg1*. (**B**) Fluorescence microscopy images of W303-1A GFP-ATG8 and W303-1A Δ*atg5* GFP-ATG8 cells after 60 min treatment with BaP1 (300 µM) or DMSO (negative control), and Rapamycin (0.2 μg/mL). Samples were collected at different time points, before (time 0) and after 30, 60, 90, and 120 min of treatment, and then observed by epifluorescence microscopy with a 100× oil immersion objective. (**C**) Effect of BaP1 or DMSO (negative control), and Rapamycin (positive control) on the Atg8p processing of W303-1A GFP-ATG8 and W303-1A Δ*atg5* GFP-ATG8 cells. Representative experiment of two independent experiments. (**D**) Quantification of cleaved GFP, values are ratios, normalized for GFP-ATG8 protein levels and PGK levels. Values are the means with SD (*n* ≥ 2). Statistical analysis was performed by two-way ANOVA. ns: non-significant, ** *p* < 0.01.

**Figure 5 jof-07-00971-f005:**
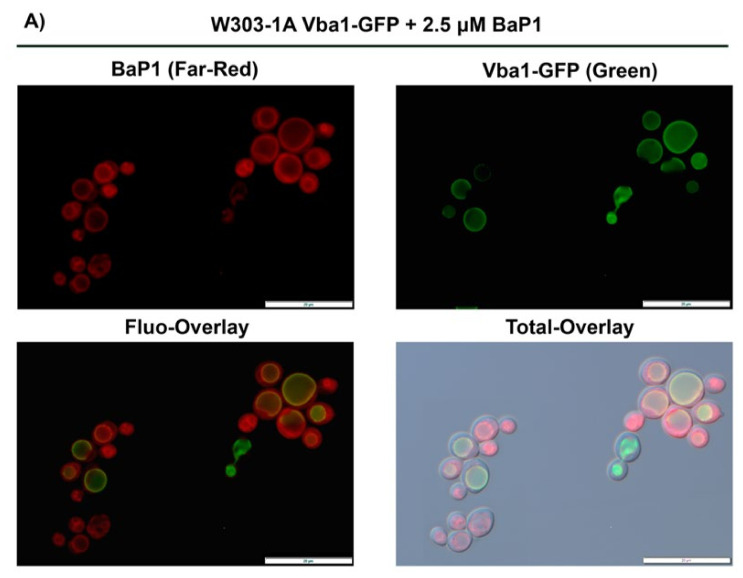
BaP1 intracellular distribution for concentrations of 2.5 μM and 300 μM. (**A**) Fluorescence microscopy images of W303-1A-pDF01-VBA1-YEGFP cells after incubation with BaP1 (2.5 µM). (**B**) Fluorescence microscopy images of BY4741-SEC66-GFP cells after treatment with BaP1 (2.5 µM). (**C**) Fluorescence microscopy images of BY4742 cells after incubation with BaP1 (300 µM). Samples were stained in PBS 1× at room temperature and visualized by epifluorescence microscopy after 5 min with a 100× oil immersion objective.

**Figure 6 jof-07-00971-f006:**
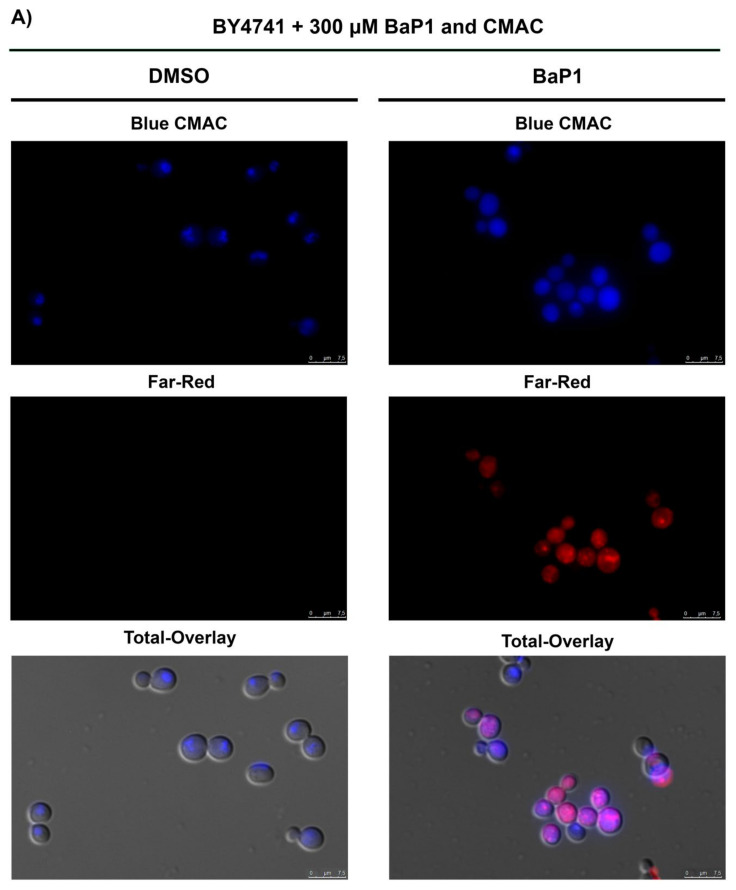
Evaluation of vacuolar membrane permeabilization and Pep4p release. (**A**) Fluorescence microscopy images of BY4741 cells after 30 min of treatment with BaP1 (300 µM) or DMSO (negative control). (**B**) Fluorescence microscopy images of W303-1A p416 ADH-pep4-EGFP cells after 30 min of treatment with BaP1 (300 µM) or DMSO (negative control). Cells were stained with CMAC to observe vacuolar permeabilization. Samples were collected at different time points, before (time 0) and after 30, 60, 90, and 120 min of treatment and then visualized by epifluorescence microscopy with a 100× oil immersion objective. (**C**) Percentage of BaP1-treated cells displaying dispersed blue staining pattern correspondent to vacuole permeabilization. The reported values are means with SD (*n* ≥ 2). At least 300 cells were counted for each condition.

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
