# Peer review of "Novel Nile Blue Analogue Stains Yeast Vacuolar Membrane, Endoplasmic Reticulum, and Lipid Droplets, Inducing Cell Death through Vacuole Membrane Permeabilization"

_jof, 2021, doi:10.3390/jof7110971_

Round 1

Reviewer 1 Report

The authors have reacted to most of my comments, I think the manuscript has improved and I can now endorse it for publication.

Reviewer 2 Report

Revised draft is OK

Reviewer 3 Report

Dear Authors, 

considering the fact that the Authors of the manuscript have revised the manuscript and exhausted my previous reservations, I believe that the manuscript may be published.

This manuscript is a resubmission of an earlier submission. The following is a list of the peer review reports and author responses from that submission.

Round 1

Reviewer 1 Report

In their work, Ferreira et al, describe a Nile Blue analogue that induces cell death. While some points the authors make are indeed convincing, others need more controls and/or additional experiments. Please find the details below:

Major points:

General: The figures are chaotic, please make sure to sort them out for a proper evaluation (Fig. 4 is missing, Fig 5 starts with B)

There is a problem with Figure 3/4: The panel B is missing in Fig3, but seems to be added instead of a Fig. 4

  1. 251: I did not find any Fig. 1F
  2. 269: I could not find any Fig. 4
  3. 262: I do not agree with the authors on the hyperpolarization. Rather, it seems that these are dead cells that accumulate the dye. To discriminate the authors have to add a PI staining to this panel. It is known that acetic acid induces cell death.

S1: Did the authors compare the same strains? It seems rather unusual that the fluorescence intensity of the untreated control differs so massively.

S2: Usually, the mean intensity is given for ROS measurement, as pure accumulation is not telling anything. The authors should go back to the raw data, gate out the PI positive population and assess the mean intensity of PI negative cells in the green channel.

  1. 282: The authors do not present flow cytometry data in S4A
  2. 284: Can the authors explain in the material and methods, how the discrimination between the cell cycle phases was done.

L 287: In this figure, the cell death by acetic acid can be seen => corresponding to my point above.

L.311: It would fit better, if the authors would include the Atg1 deletion strain in Fig.5.

Fig 5B: It seems to me that the free GFP is lower in BaP1 than in the control. Can the authors add a quantification? Further, the GFP seems to be stuck in vesicles in BaP1 treated cells, not reaching the vacuole. I would rather think that the authors see a blockage of autophagy with this substance. The quality of the micrographs is not sufficient. I can hardly see any green signal in the rapamycin-treated cells in the atg5 deletion strain. Please improve the brightness and contrast

Fig6: Please turn the subpanel letters. Further, I am not convinced by the staining and the localization. It rather seems that there is a massive crosstalk between the GFP and the dye. Could the authors please prove the single-tagged/stained controls? Otherwise, I fear this microscopy is useless. Same for C, I would like to see the single-tagged/stained controls (in all three channels) to make sure that the green channel and the blue channel do not crosstalk.

  1. 377: I cannot assess these micrographs, please ensure a better brightness/contrast. I cannot comment on CMAC distribution nor on Pep4-GFP localization. What I see are GFP-positive vesicles at the vacuolar membrane, which rather hints at a processing defect of Pep4 that at vacuolar permeabilization. Please include the potential implication of Pep4 targeting to the vacuole (including Pep1) in the discussion or perform experiments that exclude these processing/trafficking defects.

Minor points.

  1. 86 please rephrase this, the exponential phase starts directly after lag-phase when the cells start to divide exponentially. You could name it early-exponential phase
  2. 90/91: I am wondering, how the plasmids are kept without selection pressure?
  3. 106 vs. 119: I was wondering, why the PI incubation time varies between 5 and 15 min?
  4. 124: I am wondering, why H2O2 treated cells were used as control for a calcium measurement?
  5. 139: Please state, how long the rapamycin treatment was

Reviewer 2 Report

Please revise whole draft.

Reviewer 3 Report

Dear Authors, 

it was a pleasure for me to review the manuscript: " Novel Nile Blue analogue stains Yeast vacuolar membrane, endoplasmic reticulum and lipid droplets, inducing cell death through vacuole membrane permeabilization". I did not notice any factual errors. The manuscript is written in a specific way, without unnecessary descriptions. The work is understandable and constitutes a coherent whole. I believe that the manuscript is scientifically of a high standard. I have only minor comments:
Figure 2 - the charts are very reduced in size which makes their legibility poor, in particular the captions, the legend.
Figure 6 - pospidy A, B, C, vertically similar to other figures.
I would suggest looking at the bibliography because some items did not include volume and page numbers.
After these corrections, I think the manuscript will be improved and ready for publication.